# Zymography for Picogram Detection of Lipase and Esterase Activities

**DOI:** 10.3390/molecules26061542

**Published:** 2021-03-11

**Authors:** Andre Mong Jie Ng, Hongfang Zhang, Giang Kien Truc Nguyen

**Affiliations:** 1Department of Biochemistry, Yong Loo Lin School of Medicine, National University of Singapore, 8 Medical Drive, Singapore 117596, Singapore; andre.ng@u.nus.edu; 2NUS Synthetic Biology for Clinical and Technological Innovation, Centre for Life Sciences, National University of Singapore, 14 Medical Drive, Singapore 117456, Singapore; 3WIL@NUS Corporate Laboratory, Wilmar International Limited, Centre for Translational Medicine, 14 Medical Drive, Singapore 117599, Singapore; hongfang.zhang@sg.wilmar-intl.com

**Keywords:** esterase, lipase, picogram, zymogram

## Abstract

Lipases and esterases are important catalysts with wide varieties of industrial applications. Although many methods have been established for detecting their activities, a simple and sensitive approach for picogram detection of lipolytic enzyme quantity is still highly desirable. Here we report a lipase detection assay which is 1000-fold more sensitive than previously reported methods. Our assay enables the detection of as low as 5 pg and 180 pg of lipolytic activity by direct spotting and zymography, respectively. Furthermore, we demonstrated that the detection sensitivity was adjustable by varying the buffering capacity, which allows for screening of both high and low abundance lipolytic enzymes. Coupled with liquid chromatography-mass spectrometry, our method provides a useful tool for sensitive detection and identification of lipolytic enzymes.

## 1. Introduction

Lipases and esterases are ubiquitous in nature and can be found in all domains of life. These enzymes serve crucial roles for nutrition acquisition [1,2,3] in the degradation of organic materials [4,5,6] and in metabolic signalling pathways [7]. Lipases and esterases are loosely distinguished by the nature of their substrate preferences [8]. Where lipases favor the lipolysis of the ester bonds of long-chain triacylglycerols, esterases prefer shorter-chain substrates. Together, they deliver broad applications in biofuel production, cosmetic, food processing and pharmaceutical industries [9,10,11]. Therefore, it is not surprising there are ongoing intensive research efforts to screen microbes for new lipases and esterases with desirable catalytic properties for diverse industrial applications.

To facilitate the discovery of new lipases and esterases, sensitive screening methods are highly desirable and sought after. Such methods will reduce the laborious and costly experimental time and allow for detection of lipolytic enzymes presented in low abundance. A popular method for lipase screening is based on the formation of a fluorescence complex between the released free fatty acids and rhodamine B [12,13]. Although this method has been widely applied for the screening of new lipase-producing organisms, it suffers from a long incubation time and the lack of sensitivity. Other detection assays employ fluorescent substrates such as para-nitrophenyl or methylumbelliferyl-derivative substrates, which are expensive and may behave differently from natural substrates [14,15,16,17,18,19,20].

More than a decade ago, a simple activity method was described with the use of phenol red as an indicator to screen for lipases and esterases [21,22]. This detection method took advantage of the decrease in pH value, where the released free fatty acids during lipolysis creates a colour change in the indicator. As compared to other pH-based indicators, such as victoria blue, neutral red and methyl red, phenol red is the most sensitive indicator [23,24,25]. This sensitivity allows for the detection of a small changes in pH, where the method’s detection limit has been reported to be 0.5 enzyme unit (U) on a chromogenic agar plate and 5 µg amount of lipolytic enzyme by zymography [21].

Here, we present an improved lipase detection assay which is 1000-fold more sensitive than previously reported method by Singh and coworkers [21]. These improvements include the optimisation of specific buffers and experimental conditions. Under our optimised conditions, we were able to detect lipolytic activity with amounts as low as 0.01 mU (equivalent to ~5 pg of enzyme) and 0.375 mU (~180 pg) via chromogenic agar plate and zymography, respectively. To demonstrate our method’s applicability, we were able to detect pools of different lipolytic enzymes from crude and unprocessed cell culture of several microbial strains. In addition, we demonstrated that the identification of lipolytic enzymes was possible via peptide mass fingerprinting with our Zymogram-Liquid Chromatography-Mass Spectrometry (Zymogram-LC-MS) workflow. Our reported workflow will greatly facilitate the screening and discovery of new lipolytic enzymes.

## 2. Results

### 2.1. Improvement of Zymogram Detection Limit to Enable Picogram Detection of Lipases

Singh and co-workers previously reported a zymographic detection limit of 5 µg on a phenol red chromogenic agar plate [21]. In that study, the native-polyacrylamide gel-electrophoresis (native-PAGE) gel was equilibrated with 50 mM Tris-HCl buffer (pH 8.0) and overlaid with molten chromogenic agar substrate. We speculated that the strong buffering capacity would reduce the zymogram detection sensitivity. Therefore, we replaced Tris-HCl buffer with a neutral salt solution (25 mM NaCl) in our assay. The usage of 25 mM NaCl solution enhances protein stability as many proteins are known to be unstable in low ionic strength medium such as water. The NaCl concentration can also be further altered depending on the specific protein stability requirements. In addition, NaCl concentrations between 10 mM to 2 M did not change the colour of the chromogenic agar plate (data not shown).

To determine if our method would improve the zymogram sensitivity, we used commercially available enzyme Lipolase as the standard for our experiment. Different amounts of Lipolase ranging from 0.16 to 25 mU were separated by native-PAGE. After electrophoresis, the protein gels were equilibrated with either 50 mM Tris-HCl buffer (pH 8.0) or 25 mM NaCl solution at 4 °C. Equilibrated protein gels were overlaid onto a chromogenic agar plate containing 1% *v*/*v* tributyrin. After incubation at 37 °C for 30–45 min, clear yellow bands were observed on our protein gel equilibrated with 25 mM NaCl, whereas no observable yellow band was found on protein gel equilibrated with Tris-HCl buffer. The Lipolase amount, corresponding to 0.375 mU (~180 pg), was detected via our method as compared to 5 µg of Lipolase in the original method [21,26], significantly improving detection sensitivity. With an extended incubation time of 12–24 h at 37 °C, further lipolytic activities as low as 0.09 mU were observed (data not shown). Our results indicate that by carefully selecting the gel equilibrating buffer, sensitivity of the zymogram could be increased by more than 1000-fold (Figure 1). Furthermore, instead of pouring the hot molten chromogenic agar substrate, we simplified the method by overlaying the native-PAGE gel on a premade chromogenic agar plate. This change reduces the risks of thermal deactivation of enzymes and further improves the detection sensitivity. Our improved method allows for the detection of picogram quantities of lipolytic enzymes, which would otherwise gone undetected with the previously reported protocol [21]. It is noteworthy that the contrast of the gel overlay using 25 mM NaCl was not as significant as that with 50 mM Tris-HCl buffer (pH 8.0) (Figure 1). We addressed this issue by adding 0.5 mM Tris-HCl (pH 8.0) into our equilibrating buffer to improve the detection contrast of lipolytic activity against the chromogenic agar background.

### 2.2. Calibration of Lipolytic Activity Detection Limit on Chromogenic Agar Plate

To determine the sensitivity limit of the chromogenic agar plate for detection of lipolytic activity, Lipolase solution was spotted directly on the agar plate. As the assay is highly sensitive to pH and buffer strength, the existing Lipolase storage buffer was exchanged to 25 mM NaCl with a PD-10 desalting column (Cytiva, USA). Lipolase was serially diluted two-fold and spotted onto a chromogenic agar plate containing 1% *v*/*v* tributyrin, 0.01% *w*/*v* phenol red and 10 mM CaCl_2_. The release of free fatty acids upon lipolysis of tributyrin causes a colour change from red or orange to yellow. After incubation at 37 °C for 30 min, lipolytic activities with the Lipolase amounts as low as 0.01 mU were observed (Figure 2). No change in colour was observed for 25 mM NaCl solution or heat-deactivated control samples (data not shown). As the specific activity of Lipolase was estimated to be about 2000 U/mg [26], our detection limit of 0.010 mU is equivalent to about 5 pg amount of lipolytic enzyme.

### 2.3. Optimisation of Buffer Strength Effect on the Detection Sensitivity

To determine the effect of buffer strength on the detection sensitivity, tributyrin chromogenic agar plates were prepared with different concentrations of MOPS (3-(*N*-morpholino)propanesulfonic acid) (pH 7.4) ranging from 0.1 to 5 mM. Lipolase solution was spotted onto the chromogenic agar plates, incubated at 37 °C for 15 min and the yellow lipolytic zones were measured. In Table 1, our data show an inverse correlation between the lipolytic diameters zones and MOPS buffer strength. At 5 mM MOPS buffer, no visible lipolytic zone was observed at Lipolase amount below 5.6 mU, whereas at 1 mM or lower MOPS concentration zones were clearly visible at 1.4 mU of Lipolase. Therefore, by varying the buffer concentrations we could adjust the sensitivity of the detection plates, which would be useful for screening of both high and low abundance lipolytic enzymes.

### 2.4. Detection of Lipases from Crude Microbial Cultures

Having improved the sensitivity and conditions of zymographic detection, we proceeded to demonstrate the detection of several known microbial lipase-producing strains. Crude culture supernatants were separated via 12% native-PAGE. The protein gels were equilibrated with 0.5 mM Tris-HCl buffer (pH 8.0) with 25 mM NaCl for three times, at 4 °C for 10 min and overlaid on a 1% *v*/*v* tributyrin chromogenic agar plate. We anticipated that our method would allow direct detection of lipolytic activity in crude microbial culture extract without the need for concentration or lyophilisation of the samples. Figure 3 shows that lipolytic activities were detectable for all the examined strains except for *Penicillium simplicissimum*. This result supports the general applicability of our method, which facilitates the detection of lipolytic enzymes from crude and unprocessed microbial culture media.

### 2.5. Identification of Lipase with Zymogram-LC-MS

To validate the use of our method for lipase discovery, we used a combination of zymography, in-gel digestion and Liquid Chromatography-Mass Spectrometry (LC-MS) for lipase identification by peptide mass fingerprinting. Commercially available Lipolase was opted to demonstrate the proof of concept. Different concentrations of Lipolase (100, 10 and 1 mU, approximately equivalent to 50, 5 and 0.5 ng concentration of protein, respectively) were separated with a native-PAGE gel and developed on a chromogenic agar plate supplemented with 1% *v*/*v* tributyrin. Gel areas which displayed colour change from red and orange to yellow were excised and in-gel digested with trypsin (Figure 4). Resulting peptides were separated, detected, and analysed with LC-MS. For a gel piece with 100 mU Lipolase separated with native-PAGE, 180 peptides were detected, of which six peptides matched the UniProt protein sequence of *Thermomyces lanuginosus* lipase (UniProt entry O59952, LIP_THELA) (Table 2). Other proteins detected which had two or more unique peptides matches included trypsin, keratin type II cytoskeletal, RNA-directed RNA polymerase L, villin-1 and histone acetyltransferase 4 (Table 3). For samples containing 10 and 1 mU Lipolase separated with native-PAGE, peptides belonging to *Thermomyces lanuginosus* lipase were not confidently detected. Although our zymography method allows for detection of lipolytic enzyme amount as low as 0.375 mU (approximate enzyme concentration of 180 pg) (Figure 2), peptide mass fingerprinting for LC-MS analysis required approximately 50 ng of protein for reliable identification.

## 3. Discussion

Lipases and esterases are important catalysts for a wide range of industrial applications [1,2,3,4,5,7,11]. As the research for these enzymes matures, the output of commercial lipases into the market will gradually decrease. The invention of simple and sensitive detection methods for lipolytic enzymes will recatalyse the discovery of new lipases and esterases.

Here, we presented a simple, highly sensitive and economical method using phenol red as an indicator for picogram detection of lipolytic enzymes. We developed and optimised the experimental conditions to improve detection sensitivity by more than 1000-fold [21]. The key finding is that the use of neutral salt solution for equilibration greatly enhanced the detection sensitivity of lipolytic enzymes compared to a strong buffered solution, such as Tris-HCl. In addition, we used a premade chromogenic agar plate for the overlaying of the native-PAGE gel instead of pouring hot molten agar. This is extremely vital for detection of cold-adapted and heat-sensitive lipolytic enzymes, which may be deactivated at high temperatures. Using our improved protocol, we could detect lipolytic activity as low as 0.01 mU and 0.375 mU by direct spotting and zymography, respectively. Furthermore, we also demonstrated that the detection sensitivity is customizable, depending on the experimental needs, by adjusting the buffering capacity of the detection plates. In comparison with methods previously reported (Table 4), our method is easy to perform, uses inexpensive substrates, allows for quick and qualitative analyses of lipolytic enzymes which are temperature sensitive and expressed in low abundance.

The developed method was also successfully applied to unconcentrated crude microbial culture media. We were able to detect lipolytic enzymes produced by *Aspergillus aculeatus* [29], *Colletotrichum gloeosporioides* [30,31,32], *Fusarium solani* [33,34,35,36,37,38], *Penicillium expansum* [6,39,40,41,42], and *Trichoderma harzianum* [43,44,45,46,47]. The choice of substrates used in chromogenic agar detection is not limited to triglycerides, but could also extended to di- or monoglycerides, vinyl laurate and other acyl ester substrates, which would be beneficial for screening of partial-glyceride lipases or steryl esterases. In addition, by combining zymography with LC-MS, previously undiscovered lipases could be uncovered from existing or novel microbial strains.

In conclusion, we reported an improved method for picogram detection of lipase and esterase activities. We envision a broad application of our method in screening and discovery of lipolytic enzymes.

## 4. Materials and Methods

### 4.1. Isolation and Cultivation of Microbial Strains

Reagents described in all experiments were purchased from Sigma-Aldrich (Singapore) unless stated otherwise.

Microbes (*Aspergillus aculeatus*, *Colletotrichum gloeosporioides*, *Fusarium solani*, *Penicillium expansum*, *Penicillium simplicissimum* and *Trichoderma harzianum*) in this study were obtained from Wilmar International’s in-house microbial culture collection. These strains were grown on agar culture containing 1.5% agar, 1% *v*/*v* olive oil, 10 μg/mL Rhodamine B, 5% *v*/*v* Lysogeny Broth [0.5% *w*/*v* tryptone, 0.5% *w*/*v* yeast extract and 5% *w*/*v* NaCl], with minimal media [(0.073% *w*/*v* Na_2_HPO_4_, 0.035% *w*/*v* KH_2_PO_4_, 0.01% *w*/*v* MgSO_4_·7H_2_O, 0.075% *w*/*v* NH_4_NO_3_, 0.025% *w*/*v* NaHCO_3_, 0.0002% *w*/*v* MnSO_4_ and 0.002% *w*/*v* FeSO_4_] [48] and incubated at room temperature until mycelia were visually detectable.

Liquid culture used for lipase-producing enzyme contained minimal media [48] and was supplemented with 1% *v*/*v* olive oil and 5% Lysogeny Broth. Microbial strains were inoculated with sterile loops into 24-well plates (Nunc) and cultured at 28 °C, 120 rpm for 72 to 144 h.

Lipolase (lipase from *Thermomyces lanuginosus*, Sigma Aldrich L0777) was used as the lipolytic enzyme standard. Lipolytic enzymatic activity was measured using tributyrin as substrate. One-unit of activity was defined as the amount of enzyme catalysing the lipolysis of 1 µmol fatty acid per min.

### 4.2. Detection of Lipase Activity via Chromogenic Agar Plates

Chromogenic agar plates were prepared by using 0.01% *w*/*v* phenol red, 1% *v*/*v* tributyrin, 10 mM CaCl_2_ and 1% *w*/*v* agarose (Vivantis, Malaysia) as previously described [21]. The pH was adjusted to between 7.3–7.4 with 50 mM KOH using an accumet AB250 pH/ISE pH meter (Fisher Scientific, Waltham, MA, USA). Known amounts of Lipolase were spotted into chromogenic agar plates and allowed to develop at 37 °C for 5 to 60 min. Images were analysed using open-source software Fiji (ImageJ) [49]. A brightness tool was used in autoscaled mode to optimise visualisation of images (Image>Adjust>Brightness/Contrast>Auto).

### 4.3. Detection of Lipase Activity by Zymography

Native-PAGE was performed as previously described [50]. Gels were casted with 5% stacking gel and 12% resolving gel. Lipolytic enzymes were separated at 4 °C, 120 V for 60 min. After rinsing three times with distilled water, gels were equilibrated three times with 25 mM NaCl with or without 0.5 mM Tris-HCl (pH 8.0) at 4 °C for 10 min each time with gentle agitation. The inclusion of 0.5 mM Tris-HCl, (pH 8.0) in the equilibrating buffer is optional to improve the contrast of the lipolytic zones against the chromogenic agar background. Equilibrated gels were placed onto chromogenic agar plates and incubated at 37 °C for 5 to 60 min. Images were analysed using open-source software Fiji (ImageJ) [49]. A brightness tool was used in autoscaled mode to optimise visualisation of images (Image>Adjust>Brightness/Contrast>Auto).

### 4.4. Peptide Mass Fingerprinting with LC-MS

Lipolytic zones displaying colour change (from red or orange to yellow) were excised out and in-gel digested into peptides, as previously described [51]. Briefly, excised gel pieces containing potential lipolytic enzymes were destained, reduced, alkylated and tryptic digested. Extracted peptides were vacuum dried with a Concentrator plus vacuum concentrator (Eppendorf, Hamburg, Germany) and resuspended in acetonitrile and formic acid for peptide sequencing with LC-MS.

In-gel digested samples were analysed on the Ultimate 3000 High-Performance Liquid Chromatogaphy system (Thermo Scientific, Waltham, MA, USA) coupled to Q Exactive Plus Orbitrap Mass Spectrometer (Thermo Scientific, USA), as previously described [52]. 5 µL of digested protein samples were separated on the Acclaim PepMap RSLC C18 Column (300 µm × 150 mm, 3 µm, 100 Å) (Thermo Scientific, USA). Flow rate was set at 10 µl/min, with 0.1% *v*/*v* formic acid (Merck, Kenilworth, NJ, USA) in distilled water as solvent A and 0.1% *v*/*v* formic acid in acetonitrile (Merck, USA) as solvent B. The running gradient was as followed: 5% solvent B from 0 to 2 min, 5 to 70% solvent B from 2 to 21 min, 70 to 99% solvent B from 21 to 21.3 min, 99% solvent B from 21.3 to 25.5 min, 99 to 1% solvent B from 25.5 to 26 min, and 1% solvent B from 26 to 30 min. Data acquired on the mass spectrometer was operated in full scan mode followed by data-dependent tandem mass spectrometry (MS/MS) acquisition over a mass range of 300–1800 *m/z*.

### 4.5. Data Analysis

Data analysis was performed with PEAKS Studio 7.5 (build 20150615) as previously described [53]. Sequenced peptides were identified, and de novo assembled. Assembled peptides were searched against UniProt reference proteome database (559 213 sequences, updated on 12 September 2019) with a mass error tolerance for parent ion mass was ±10 ppm with fragment ion as ±0.5 Da. Methionine oxidation and protein *N*-terminal acetylation were chosen as variable modifications, and cysteine alkylation by iodoacetamide was chosen as a fixed modification. The protease was specified as trypsin with three maximum missing cleavages.

## Figures and Tables

**Figure 1 molecules-26-01542-f001:**
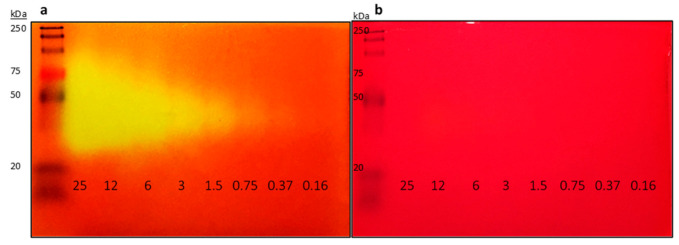
Native-PAGE (12%) and zymogram analysis of Lipolase on 1% *v*/*v* tributyrin chromogenic agar plates. (**a**) Improved detection using 25 mM NaCl as our equilibration buffer. (**b**) Original detection method using 50 mM Tris-HCl (pH 8.0) as the equilibration buffer [21]. Lipolase amounts from 25 to 0.16 mU were separated at 4 °C, 120 V for 60 min. Lipolytic activity is indicated by a colour change from red or orange, to yellow.

**Figure 2 molecules-26-01542-f002:**
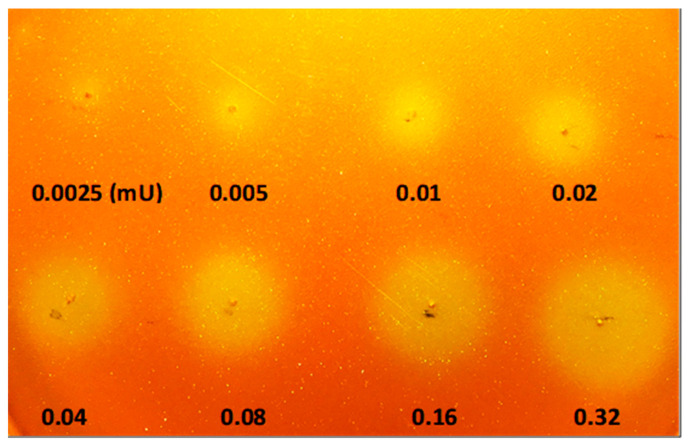
Detection of lipolytic activity on chromogenic agar plate supplemented with 1% *v*/*v* tributyrin. Lipolase amount corresponding to 0.32, 0.16, 0.08, 0.04, 0.02, 0.01, 0.005, and 0.0025 mU were spotted on the plate at 37 °C for 15 min. Lipolytic activity is indicated by a colour change from red or orange to yellow.

**Figure 3 molecules-26-01542-f003:**
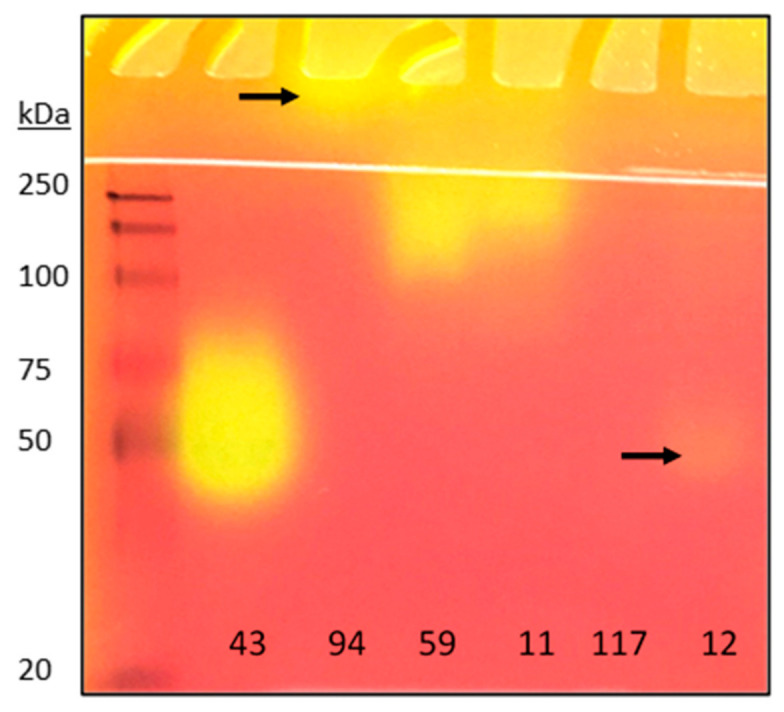
Zymogram analysis of crude microbial cultures. Lipolytic activity is indicated by a colour change from red or orange to yellow. Lipase(s) in strain 94 did not migrate down (indicated by the arrow), probably due to their basic nature. Lipase in strain 12 was expressed at low amount and indicated by the arrow. Microbial strains used were 43 = *Aspergillus aculeatus;* 94 *= Penicillium expansum,* 59 *= Fusarium solani,* 12 *= Colletotrichum gloeosporioides;* 117 *= Penicillium simplicissimum;* 11 = *Trichoderma harzianum.* They were numbered according to our Wilmar International’s in-house microbial culture collection.

**Figure 4 molecules-26-01542-f004:**
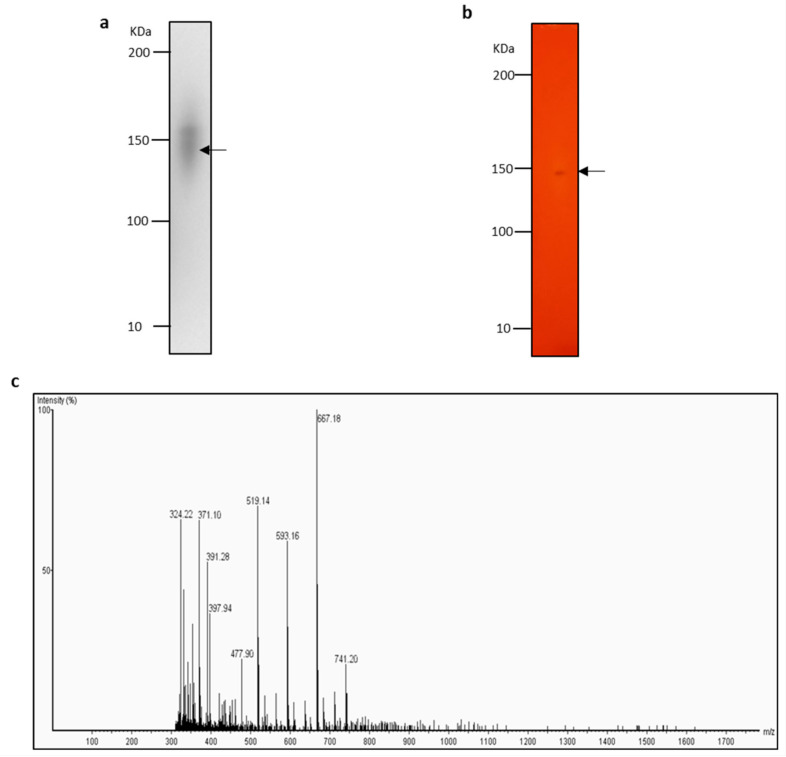
Identification of Lipolase by Zymogram Liquid Chromatography-Mass Spectrometry (LC-MS). (**a**) Coomassie blue-stained native-PAGE gel of 100 mU Lipolase. The area marked with an arrow corresponds to excised area on zymogram. (**b**) Zymogram of native-PAGE gel of 100 mU Lipolase. The area marked with an arrow was excised, in-gel tryptic digested and analysed with LC-MS. (**c**) Total ion chromatogram of peptides detected from Zymography-LC-MS of excised gel piece in Figure 4b.

**Table 1 molecules-26-01542-t001:** Diameter of lipolytic zones (mm) in 1% *v*/*v* tributyrin chromogenic agar plates.

Lipolase Amount (mU)	MOPS Concentration (mM)				
	5	1	0.5	0.25	0.1
180	7	9	11.5	12.5	15
90	6	8	11	11.5	14
45	4.5	7	9.5	11	13
22.5	5	6	9	10	12
11.2	4.5	6	8.5	9	11
5.6	3	5.5	8	8.5	10
2.8	- *	5	7	8	9.5
1.4	-	4	6	7	8.5

* Indicates no detectable lipolytic zones. Chromogenic agar plates were spotted and incubated at 37 °C, for 15 min. MOPS: 3-(*N*-morpholino)propanesulfonic acid.

**Table 2 molecules-26-01542-t002:** List of identified peptides matches to in-gel digested piece containing 100 mU Lipolase.

No.	Accession (UniProt)	Mass	Retention Time	Peptide
1	O59952|LIP_THELA	760.4079	8.33	SVADTLR
2		1164.625	8.67	ITHTNDIVPR
3		928.5342	9.61	SGTLVPVTR
4		2192.069	11.23	PPREFGYSHSSPEYWIK
5		1630.7263	12.8	GN(+98)GYDIDVFSYGAPR
6		1661.841	14.57	SIENWIGNLNFDLK

**Table 3 molecules-26-01542-t003:** List of proteins identified from in-gel digested 100 mU Lipolase with more than two unique peptide coverages.

No.	Accession (UniProt)	Coverage (%)	No. of Peptides	No. of Unique Peptides	Protein Description
1	P00761|TRYP_PIG	44	20	16	Trypsin
2	A5A6M6|K2C1_PANTR	15	7	5	Keratin type II cytoskeletal 1
3	P04264|K2C1_HUMAN	15	7	5	Keratin type II cytoskeletal 1
4	O59952|LIP_THELA	25	6	6	Lipase
5	Q6XQI4|L_JUNIN	1	2	2	RNA-directed RNA polymerase L
6	Q0DKN3|VLN1_ORYSJ	7	2	2	Villin-1
7	Q9LG11|HAC4_ARATH	4	2	2	Histone acetyltransferase 4

**Table 4 molecules-26-01542-t004:** Comparison of several zymography detection methods for lipases and esterases.

Detection Method	Detection Principle	Detection Limit	Assay Time (min)	Advantages	Disadvantages	Reference
Zymography-LC-MS	pH changes from hydrolysis of triacylglycerols	0.375 mU or 180 pg	5–15	Low cost Rapid Detect picogram amounts Versatile with fatty substrates Identify protein sequences Adapted for temperature-sensitive proteins	Coupled with LC-MS detection	(This study)
Zymography	White precipitate from esterification and transesterification	Not stated	1440	Detection of esterification	Long incubation time Low sensitivity	[27]
Zymography	White precipitate from esterification of alcohol and fatty acid	Not stated	30–720	Detection of esterification Low cost	Long incubation time Low sensitivity	[28]
Zymography	pH changes from hydrolysis of triacylglycerols	0.5–2.0 U or 5–20 µg	5–15	Low cost Rapid	Strong buffering capacity Unable to detect picogram amounts Molten agar overlay Denaturation of proteins Low sensitivity	[21]
Zymography	pH changes from hydrolysis of triacylglycerols	0.17–0.66 U	360–720	Low cost	Long incubation time Low sensitivity	[22]

## Data Availability

Data set presented in this study is available in this article.

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
