# Peer review of "Zymography for Picogram Detection of Lipase and Esterase Activities"

_molecules, 2021, doi:10.3390/molecules26061542_

Round 1

Reviewer 1 Report

The authors resort to one method already presented in “Appl Microbiol Biotechnol 2006 May;70(6):679-82., A simple activity staining protocol for lipases and esterases” and modify it slightly to enhance the detection limit. This is valuable, but in this paper, I cannot see the justification.

The authors justify this upgrade by coupling with LC-MS and claim and quote “ Coupled with LC-MS our method provides a useful tool for sensitive detection and identification of lipolytic enzymes. “. Although in lines 238, 239 and 240 authors say and quote “Although our zymography method allows for detection of lipolytic enzyme 238 amount as low as 0.375 mU (approximate enzyme concentration of 180 pg) (Figure 2), 239 peptide mass fingerprinting for LC-MS analysis required approximately 50 ng of protein 240 for reliable identification” This means that instead of 0.375 mU it must be used 277 mU. So, what´s the use of a higher sensitivity??

Other issues:

  • The Results section is somewhat confusing.
  • All around the text the authors change the way of presentation of enzyme units (sometimes mU, sometimes µU)
  • Nothing is said about the detection limits of other detection methods, mentioned in the introduction, that using calibration curves can usually enable the quantification in a very short time also.
  • Nothing is said about how authors quantify the halo of stained phenol red if the concentration is not known.
  • Before using LC-MS it is necessary a sample preparation (lines 108-110). And what about preparing the initial samples and using them directly in LC-MS without using Zymography first?
  • In 3.1 there is a bit confused about using Tris-HCl or NaCl, see lines 165-167.
  • Why using MOPs to study the buffer strength?

Author Response

Response to Reviewer 1’s Comments

Comment 1

The authors resort to one method already presented in “Appl Microbiol Biotechnol 2006May;70(6):679-82., A simple activity staining protocol for lipases and esterases” and modify it slightly to enhance the detection limit. This is valuable, but in this paper, I cannot see the justification.

The authors justify this upgrade by coupling with LC-MS and claim and quote “Coupled with LC-MS our method provides a useful tool for sensitive detection and identification of lipolytic enzymes. “. Although in lines 238, 239 and 240 authors say and quote “Although our zymography method allows for detection of lipolytic enzyme 238 amount as low as 0.375 mU (approximate enzyme concentration of 180 pg) (Figure 2), 239 peptide mass fingerprinting for LC-MS analysis required approximately 50 ng of protein 240 for reliable identification” This means that instead of 0.375 mU it must be used 277 mU. So, what´s the use of a higher sensitivity??

Response to Comment 1

The crude microbial cultures are generally very complex with hundreds or thousands of unidentified proteins. To purify and discover a new lipase would require multiple steps of chromatography and fractionations. Lipases in unconcentrated fraction is often very low and thus a highly sensitive method is needed to quickly determine which fractions contain the desired lipolytic activity.

These fractions can be pooled and concentrated for protein identification by mass spectrometry, which would require about 50 ng for reliable analyses due to limitation of the MS instrument. Previous zymographic method only allow the detection of up to 5 microgram concentration of lipases.

Our improved detection method addresses this limitation by facilitating the detection of lipolytic activity of as low as 0.375 mU, which is approximately 180 pg of purified enzyme. This significant improvement will facilitate the detection of lipolytic enzymes which was overlooked by other groups in the past. We believe our highly sensitive detection method will open new horizons for discovery of new lipolytic enzymes.

Comment 2

Other issues:

The Results section is somewhat confusing.

All around the text the authors change the way of presentation of enzyme units (sometimes mU, sometimes μU)

Response to Comment 2

Comment considered, all enzymatic units have been changed and standardised to mU.

Comment 3

Nothing is said about the detection limits of other detection methods, mentioned in the introduction, that using calibration curves can usually enable the quantification in a very short time also.

Response to Comment 3

The main application of our method is for zymographic determination of lipases from complex microbial cultures, which may contain thousands of unidentified proteins.  Previous method required 5 micrograms of lipolytic protein for zymogram, which we improved the detection sensitivity by 1000-fold. Our highly sensitive method will allow the quick identification of lipolytic target protein band from protein gels containing complex mixtures of cellular proteins.

Comment 4

Nothing is said about how authors quantify the halo of stained phenol red if the concentration is not known.

Response to Comment 4

Our method is semi-quantitative, and we had shown the halo development is correlated to the predetermined amount of Lipolase concentration spotted into the agar. If there is a microbial culture with unknown enzyme concentration, a standard curve using commercial lipase could be easily replicated to qualitatively estimate the total lipolytic activity.

Comment 5

Before using LC-MS it is necessary a sample preparation (lines 108-110). And what about preparing the initial samples and using them directly in LC-MS without using Zymography first?

Response to Comment 5

As mentioned earlier, crude microbial cultures are complex with hundreds or thousands of unidentified proteins. Zymogram help to identify the target protein band and served as an enrichment or semi-purification step to reduce the background noise for protein identification.

Zymography will help to filter away undesired, non-lipolytic proteins which would affect the detection sensitivity of LCMS.

Comment 6

In 3.1 there is a bit confused about using Tris-HCl or NaCl, see lines 165-167.

Response to Comment 6

Figure 2 highlights the detection sensitivity of our method (a) against the original method (b). By just using 50 mM NaCl as our equilibration buffer, Lipolase amount of 0.16 mU could be detected after 60 min of incubation.

Comment 7

Why using MOPs to study the buffer strength?

Response to Comment 7

MOPS have a pKa of 7.2, very close to the colour changing range of phenol red, and thus it fit the purpose of our study.

Reviewer 2 Report

The article entitled Zymography for Picogram Detection of Lipase and Esterase Activities presents an improved zymography to detect picograms of lipase activity. The work pretends to provide an easy, low-cost and sensitive methodology to screen and discover new lipase activities from raw lipolytic extracts. In my opinion, the methodology and the results analysis are non-accurately conducted and must be improved. The manuscript has several English mistakes and must be reviewed.

In particular I will stress the following recommendations:

Methods:

In 2.1. Please correct the name of the used fungi, they must be with italic letters (Aspergillus aculeatus, Colletotrichum gloeosporioides, Fusarium solani, Penicillium expansum, Penicillium simplicissimum, Trichoderma harzianum).

Authors employed tributyrin as substrate to measure lipase activity. Tributyrin is a water soluble triglyceride, then, several esterases show activity towards this substrate. Instead this substrate, insoluble triglycerides possessing more than 6 carbon atoms (triolein, trioctanoin, etc) are more suitable ones to detect exclusively lipase activity.

Authors employed an Apple iPhone 8 camera in auto-mode in order to take photographs of the zymograms. In my opinion, equipment that is more accurate must be employed in order to properly imaging and processing the obtained images (like GelDoc Imaging system, Bio-rad). If authors do not have this equipment, they must process images employing imaging software as Fiji or Zen. The proper imaging of the gels will determine the quantitative limits of the detection method.

Results:

Figures 1 and 2 must be renumbered since the manuscript starts describing figure number 2, so the figure number 1 appears later than the figure 2, this does not have any sense.

Figure 2 must be improved, please add the molecular weight of your standard marker in order to properly visualize the molecular weight of the tested Lipolase.

In the section “3.2. Detection limit of lipolytic activity on chromogenic agar plate” the results might be improved if authors employ an imaging software in order to quantify the developed yellow color that they have in their plates. With these numbers, they can include a calibration curve of the lipase activity or lipase ammount versus the developed yellow coloration.

In page 6, line 205, Table 1 does not correspond with the provided description. Please correct.

Figure 3 must be corrected. The molecular weight marker again lack in the molecular weight of the bands. Again the lane labels are incorrect, so please do they correctly match with their respective lane. As well, please include the molecular weight of the tested lipases.

Discussion:

I would include a comparative table were other lipase detection methods based on zymography are employed. In this table highlight the advantages and disadvantages of the current developed method compared with the state-of-the-art ones. To this aim, I would include the following data: sensitivity, assay time, cost, detection principle and limitations and advangates.

Author Response

Response to Reviewer 2’s Comments

The article entitled Zymography for Picogram Detection of Lipase and Esterase Activities presents an improved zymography to detect picograms of lipase activity. The work pretends to provide an easy, low-cost and sensitive methodology to screen and discover new lipase activities from raw lipolytic extracts. In my opinion, the methodology and the results analysis are non-accurately conducted and must be improved. The manuscript has several English mistakes and must be reviewed.

In particular I will stress the following recommendations:

Comment 1

In 2.1. Please correct the name of the used fungi, they must be with italic letters (Aspergillus aculeatus, Colletotrichum gloeosporioides, Fusarium solani, Penicillium expansum, Penicillium simplicissimum, Trichoderma harzianum)

Response to Comment 1

Revised. Formatting error occurred during free-format submission.

Comment 2

Authors employed tributyrin as substrate to measure lipase activity. Tributyrin is a water soluble triglyceride, then, several esterases show activity towards this substrate. Instead this substrate, insoluble triglycerides possessing more than 6 carbon atoms (triolein, trioctanoin, etc) are more suitable ones to detect exclusively lipase activity.

Response to Comment 2

Tributyrin has limited solubility in water. Butyric acid, the hydrolysed product of tributyrin, is water soluble. Tributyrin forms an immiscible layer upon addition to water, which would form a hydrophobic phase to induce lid opening for lipases. The use of tributyrin enables detection of both esterases and lipases. We do agree that to detect exclusively lipase, longer chain triglycerides should be employed.

Our method is meant to be a dynamic detection system where besides tributyrin, other fatty substrates such as the longer chain TAGs, DAGs or MAGs, such as you have mentioned, and can be easily added as components into the detection mixture for zymography.

Comment 3

Authors employed an Apple iPhone 8 camera in auto-mode in order to take photographs of the zymograms. In my opinion, equipment that is more accurate must be employed in order to properly imaging and processing the obtained images (like GelDoc Imaging system, Bio-rad). If authors do not have this equipment, they must process images employing imaging software as Fiji or Zen. The proper imaging of the gels will determine the quantitative limits of the detection method.

Response to Comment 3

Agree with your suggestion. We had processed our figures using software as recommended.

Comment 4

Figures 1 and 2 must be renumbered since the manuscript starts describing figure number 2, so the figure number 1 appears later than the figure 2, this does not have any sense.

Response to Comment 4

Revised. Formatting error occurred during free-format submission.

Comment 5

Figure 2 must be improved, please add the molecular weight of your standard marker in order to properly visualize the

Response to Comment 5

Revised as per comment. However, for Native-PAGE, molecular markers do not accurately describe the molecular weight of tested lipolytic enzymes. However, it can serve as a reference for protein migration.

Comment 6

In the section “3.2. Detection limit of lipolytic activity on chromogenic agar plate” the results might be improved if authors employ an imaging software in order to quantify the developed yellow color that they have in their plates. With these numbers, they can include a calibration curve of the lipase activity or lipase ammount versus the developed yellow coloration.

Response to Comment 6

We did not provide a calibration curve for 2 reasons. First, while there is correlation between enzyme concentration and yellow coloration, the halo diameter varies with incubation time. Second, our assay is sensitive to pH, and the pH of the microbial culture media vary significantly among different strains. We often compared between heat-treated culture and non-treated sample to eliminate the background or the colour change due to media. It is thus challenging to provide an accurate calibration curve to estimate the amount of lipase in crude microbial cultures.

Comment 7

In page 6, line 205, Table 1 does not correspond with the provided description. Please correct.

Response to Comment 7

Revised. Formatting error occurred during free-format submission.

Comment 8

Figure 3 must be corrected. The molecular weight marker again lack in the molecular weight of the bands. Again the lane labels are incorrect, so please do they correctly match with their respective lane. As well, please include the molecular weight of the tested lipases.

Response to Comment 8

Revised as per comment. Formatting error occurred during free-format submission.

Comment 9

I would include a comparative table were other lipase detection methods based on zymography are employed. In this table highlight the advantages and disadvantages of the current developed method compared with the state-of-the-art ones. To this aim, I would include the following data: sensitivity, assay time, cost, detection principle and limitations and advangates.

Response to Comment 9

Revised as per comment. Thank you for your helpful suggestion

Round 2

Reviewer 1 Report

The paper can be accepted in the present form.

Author Response

Noted. Thank you for your helpful comments to make the manuscript better.

Reviewer 2 Report

Authors addressed almost all the suggested changes. However they said that analysed their images with suggested software and I did not see any description of these analysis in the methodology. Please incorporate this information. 

Author Response

Noted, revised as per comment.

Thank you for your helpful comments to make the manuscript better.